# World Trends in Dental Ergonomics Research: A Bibliometric Analysis

**DOI:** 10.3390/ijerph21040493

**Published:** 2024-04-17

**Authors:** Wita Anggraini, Dewi Ranggaini, Annisaa Putri Ariyani, Indrani Sulistyowati

**Affiliations:** 1Anatomy Subsection, Department of Oral Biology, Faculty of Dentistry, Universitas Trisakti, Jakarta 11440, Indonesia; annisaa.ariyani@trisakti.ac.id (A.P.A.); indrani@trisakti.ac.id (I.S.); 2Physiology Subsection, Department of Oral Biology, Faculty of Dentistry, Universitas Trisakti, Jakarta 11440, Indonesia; monica.dewi.r@trisakti.ac.id

**Keywords:** bibliometric analysis, ergonomic, dentist, musculoskeletal disorders

## Abstract

Dental ergonomics provides an overview of dentists’ work efficiency. The objective of this study was to obtain quantitative information and produce a visualization of the network of scientific publications on the topic of ergonomics and dentistry using bibliometric analysis. Data mining was conducted using the Scopus database and Boolean expressions (ergonom* AND dentist*) on 14 April 2023. Data extraction and analysis were performed using Open Refine version 3.5.2., VOSviewer version 1.6.17., VOSviewer thesaurus, Microsoft Excel, and Tableau Professional version 2020.1.2. A total of 682 documents were identified, with the United States having the largest number of documents and citations (89 documents, 1321 citations). *Work*, *Dentistry Today*, and the *International Journal of Environmental Research and Public Health* were the top three sources. Ergonomics and musculoskeletal disorders (MSDs) are two of the very prominent keywords, with research topics covering prevalence, causes, factors related to causes, prevention, assessment, rehabilitation, evaluation, and intervention. There was no research on ergonomic interventions that collaborated with human factors and ergonomics (HFE). We conclude that the trending topic of dental ergonomics research topics around the world is centered on MSDs. The future research challenge is to apply HFE science to improve the health, safety, efficiency, and quality of dentists’ work.

## 1. Introduction

The focus of ergonomics is on humans and their interactions with products, equipment, facilities, procedures, and the environment in which humans live and work every day. Ergonomics seeks to prevent work-related musculoskeletal disorders through identifying, evaluating, and controlling the risk factors in the physical workplace [1]. The dental profession involves repetitive movements combined with forceful movements, awkward postures, and inadequate recovery time. This profession requires precision and has high visual requirements due to the narrow working area, i.e., the oral cavity. Therefore, dentists are at high risk of developing work-related musculoskeletal disorders (WMSDs) [2]. WMSDs are a subcategory of musculoskeletal disorders (MSDs). The Occupational Safety and Health Association (OSHA) defines musculoskeletal disorders (MSDs) as disorders of the skeletal muscles, nerves, tendons, ligaments, joints, cartilage, or vertebral discs, which occur slowly over time due to repeated wear and tear or microtrauma [3,4].

WMSDs are a significant problem for dentists and have been discovered early in their careers, even among dental students. Yusof et al., in their research, found that there were significant differences in posture among clinical students in their third, fourth, and fifth years of study. Body pain and the development of MSDs in fourth-year students were caused by poor posture in the legs, while in fifth-year students, they were caused by poor posture in the forearms. Students tend to work in awkward postures that are dangerous, because they are inexperienced and prioritize their patients. These conditions are exacerbated by high levels of pressure to complete treatments on time [5].

The prevalence of MSDs in dentists throughout the world varies from 10.8% to 97.9% [6]. In their research, Batham and Yasobant found that 92.7% of dentists had experienced MSDs during the last 12 months, and in the last 7 days, 84.7% of dentists had complaints of MSDs [7]. The most common MSDs in dentists are back pain, followed by neck pain, shoulder pain, high tension of the trapezius muscle, tendinitis, carpal tunnel syndrome, pinched nerves, early arthrosis, myopia, and auditive changes [8,9]. Pain in the muscles is an alarm in the body before the risk of paralysis and injury occurs, which has the potential to end a career early [10]. In their systematic review, Bret and Gorce reported that the highest prevalence of MSDs in dentists was in the lower back (>60%), shoulders, and upper extremities (35–55%). The main cause was an awkward posture repeated over a long time [11]. Soo et al. reported that dentists’ susceptibility to MSDs reached 68% to 100% in various parts of the body, especially in the neck (26–92%), shoulders (25–92, 7%), and lower back (29% to 94.6%). The causes and problems of MSDs are multifactorial; key risk factors are female dentists (57.1%), awkward posture (50%), long periods of work (30%), and specialist dentists (42.9%) [12]. To maximize the performance of dentists, human factors and ergonomics (HFE) is a special study area aimed at improving the health, safety, efficiency, and quality of dentists’ work while also having a positive impact on patient safety. HFE interventions in health services are categorized into the following: (1) physical ergonomic interventions, (2) cognitive ergonomic interventions, and (3) organizational ergonomic interventions [13].

Scientific publications on ergonomics related to dentists have been around since the 1960s and have consistently increased from year to year. The research subjects are not only dentists but also dental assistants [14], dental hygienists [15], and dental students [16]. The scope of the research is very broad, including the work environment [17,18] and ergonomic interventions [16,19,20,21,22]. These research articles have contributed to producing dental ergonomic principles, so that dentists always work with an “ergonomic culture” [23], and provide strategies for preventing MSDs [24,25].

Based on this background, a bibliometric analysis was carried out on the topic of ergonomics and dentists. Bibliometric analysis is a statistical tool for mapping the highest and current levels of scientific development and identifying research gaps and trends for various purposes, such as searching for research opportunities and supporting scientific research [26]. In bibliometric analysis, VOSviewer software 1.6.17 (Centre for Science and Technology Studies, Leiden, The Netherlands) may be used to create a publication network data map accompanied by data visualization that includes co-authorship, co-occurrence, citation, bibliographic coupling, and co-citation links. The use of this software helps researchers, librarians, and publication database managers to obtain a network map of scientific publications including authors or researchers, organizations, countries, and keywords [27,28,29]. VOSviewer users can import bibliographic databases from Scopus, PubMed, or Web of Science [30]. Bibliometric analysis techniques have developed over time and continuously, to measure the impact of publishing articles within the scientific community. All data are presented in the form of mapping to describe the relationships between nodes in the expanded analysis [31].

The aim of this research was to obtain quantitative information and visual information on world trends in ergonomics research related to dentists or dental ergonomics research in all Scopus-indexed publications up to 2023. The analyses in this research included performance analysis, an analysis of journals and articles, an analysis of collaboration between authors and between countries, and an analysis of the intellectual structure of authorship, which maps publication countries, sources, authors, citation networks, and co-citation networks between authors.

## 2. Materials and Methods

This research was carried out in two stages. The first stage was an exploratory stage of searching for research topics using several keywords with Boolean expressions in the Scopus database. The purpose of this preliminary research was to find research topics with keywords that could provide the most data information (Table 1). A preliminary research topic search was carried out on 5 April 2022 using Boolean sentences in the Scopus electronic database. To search for phrases in Scopus, double quotes are used (“), wildcards (*), and Boolean operators (OR, AND, NOT). The purpose of double quotes is to tell Scopus that these are “loose phrases”, meaning that the words must be together. The use of wildcards (*) is to represent a number of characters, and Boolean operators are used to expand or narrow search parameters when using databases or search engines. The default search field in Scopus uses TITLE-ABS-KEY because the Scopus database is an abstract indexer only [32,33].

Based on the results of data mining with several topics in our preliminary research, the topics of ergonomics and dentistry were selected, which provided the largest number of documents. During this research, data mining was carried out on the Scopus database using Boolean expressions (ergonom* AND dentist*) on 14 April 2023. The search results were exported into a Comma-Separated Value (CSV) file in Microsoft Excel [34]. Microsoft Excel software was also used to analyze all information from recruited sources. To work on visualization and bibliometric construction, VOSviewer software version 1.6.17 was used, and to clean the data, Thesaurus_text in VOSviewer and Open Refine software version 3.5.2 were used.

In the thesaurus step, keywords that have the same meaning (synonyms/hyponyms) were combined or deleted. The document distribution visualization tool was Tableau Professional software version 2020.1.2 (Salesforce Inc., Singapore). The bibliography analysis attributes in VOSviewer software 1.6.17 include co-authorship, co-occurrence, citation, bibliography coupling, co-citation of authors, organizations, and countries [35]. The bibliometric analysis flow can be seen in Figure 1.

## 3. Results

### 3.1. Performance Analysis

#### 3.1.1. Publication Frequency by Year

There were 682 research publications on ergonomics and dentistry in English-language journals from 1965 to 2023 in the Scopus database. Figure 2 shows the decrease in the number of publications, covering the years 1979 (1 publication), 1985 (2 publications), 1992 (2 publications), 1993 (2 publications), 1994 (2 publications), 1995 (3 publications), and 1997 (3 publications). Since 2020, publications on dental ergonomics have increased sharply, and they reached a peak in 2021 of 43 publications.

#### 3.1.2. Contribution of Countries to the Field of Dental Ergonomics

A bibliometric coupling analysis was used to evaluate the number of articles published based on the authors’ country of origin (Figure 3). There were 90 countries with at least one article and zero citations. The United States occupied the highest position for ergonomic dentistry publications with 89 articles. The ranking of countries based on the number of articles can be seen in Table 2.

#### 3.1.3. Number of Article Citations by Country

Citation analysis was carried out on country analysis units with a maximum limit of 25 countries in one article, a minimum of one article, and one citation per country (Table 3). The results of this analysis were that out of 90 countries, 69 met these limits. Of the 20 countries with the highest citation weight, the United States was again in first place with 1321 citations for its 89 articles. The exciting thing was that Greece, with just three documents, obtained 287 citations and was ranked sixth.

### 3.2. Analysis of the Source

#### 3.2.1. Source Analysis Based on the Number of Documents

Source analysis was performed using VOSviewer 1.6.17 with bibliographic coupling. This was based on the number of documents or articles and carried out on the sources unit, with a threshold of each journal having at least one article and zero citations. The aim of providing zero citations is that all sources can be presented through this application (Figure 4). The results showed that 323 Scopus-indexed sources published the 682 articles obtained via data mining, and the most extensive collection of connected sources consisted of 174 articles. In Table 4, it can be seen that *Work* was the top source, with 19 articles, followed by *Dentistry Today* (18 articles), the *International Journal of Environmental Research and Public Health* (14 articles), and the *Journal of The American Dental Association* (12 articles).

#### 3.2.2. Citation Analysis of Sources

This citation analysis used VOSviewer in the sources unit, with a minimum threshold of having one document and zero citations. The aim of providing zero citations is that all sources can be presented through this application (Figure 5). As a result of taking this approach, 323 sources indexed by Scopus were recruited, and the largest collection of connected sources consisted of 162 sources. Table 5 shows the results of an analysis of sources based on citations, where the *Journal of The American Dental Association* is the top source with a citation weight of 460, followed by *BMC Musculoskeletal Disorders* (421 citations), *Work* (338 citations), and *Applied Ergonomics* (262 citations).

### 3.3. Analysis of the Article

Our analysis of articles or documents aimed to discover which articles have had the greatest influence on research trends in relation to the topics ‘ergonomics’ and ‘dentists’ or, in other words, research trends in the field of dental ergonomics. For the analysis, VOSviewer was used with citation analysis as the type and analysis documents as the unit. The lower citation threshold was zero, which we used to obtain all article data in this research. VOSviewer displayed 682 articles and their information, including the authors’ names, titles, source information (source name, volume, issue, page), and year of publication.

The article ‘Prevalence of musculoskeletal disorders in dentists’ published in the journal *BMC Musculoskeletal Disorders* in 2004 occupied the top citation position. Alexopoulos E.C., Charizani F., and Stathi I.C. wrote this article (Figure 6). The significant citation weight for the article shows the authors’ enormous contribution to the development of dental ergonomics. In Table 6, the 15 most frequently cited articles are shown.

### 3.4. Analysis of Co-Authorship

#### 3.4.1. Co-Authorship between Authors

Co-authorship analysis examines interactions between authors in any scientific field. Co-authorship is a formal arena for collaboration between writers and experts and can even occur across and between scientific fields [36]. Co-authorship analyses in the author analysis unit from 1733 authors identified 42 with a minimum of four articles [37]. Next, an analysis was carried out using Microsoft Excel (Microsoft, Redmond, WA, USA). In Figure 7, based on the network visualization, three clusters have an extensive network, namely, Clusters 1, 2, and 3. The exciting thing is that the authors from Cluster 1 and Cluster 2, apart from writing together with authors in their cluster, are also shown to collaborate between the clusters.

The joint writing collaboration between Clusters 1 and 3 shows collaboration in scientific fields and institutions. From Clusters 1 and 3, eight authors were found from Social Medicine and Environmental Medicine, the Institute of Occupational Medicine, Goethe-University (Germany), namely, Groneberg D.A., Ohlendorf D., Holzgreve F., Wanke E.M., Fraeulin L. Maurer-Grubinger C., Hauck I., and Nowak J. In addition, from Cluster 1, several different author affiliations were found including one author from the Medical Center of the Johannes Gutenberg, Department of Orthodontics, University Mainz (Germany), namely, Erbe C.; one author from the Principles of Prevention and Rehabilitation Department (GPR), Institute for Statutory Accident Insurance and Prevention in the Health and Welfare Services (Germany), namely, Nienhaus A.; and one author from the Department of Dental Radiology, Institute of Dentistry, Goethe-University (Germany), namely, Betz W. In Cluster 3, two authors were found from the Institute for Occupational Health and Safety (IFA)—German Social Accident Insurance (DGUV), Germany, namely, Ditchen D. and Hermanns I.

Joint authorship in Cluster 2 also demonstrates the degree of collaboration between scientific fields and institutions based on author affiliation. From Cluster 2, one author was found from the Mechatronics Department, Polytechnic University (Romania), namely, Argesanu V.; one author from the Ergonomics Department, Faculty of Dental Medicine, Victor Babes University of Medicine and Pharmacy (Romania), namely, Anghel M.D.; one author from the Department of Mechanical Machinery, Equipment, and Transport, Polytechnic University of Timisoara (Romania); and two authors from the Department of Periodontology, Faculty of Dental Medicine, Victor Babes University of Medicine and Pharmacy (Romania), namely, Stratul S. and Rusu D.

#### 3.4.2. Co-Authorship between Countries

Co-authorship analysis was carried out on country analysis units with a maximum limit of 25 countries in one article and a minimum of one article per country, with a zero-citation limit. We found that 90 countries met these limits, and the most extensive set comprised 35 connected countries (Figure 8). In Table 7, the ten countries with the top co-authorship are presented.

### 3.5. Analysis of the Intellectual Structure

Our intellectual structure analysis aimed to determine which authors, articles, or sources have had the most significant influences on ergonomic dentistry research trends [38].

#### 3.5.1. Analysis of the Authors’ Keywords

The aim of analyzing the authors’ keywords is to find the correlation between keywords and the articles’ topics, in this case, so that readers will find it easy to search for various dimensions of research on the themes of ‘ergonomics’ and ‘dentistry’. The analysis was performed using the VOSviewer application, namely, a co-occurrence analysis of the authors’ keywords with a minimum threshold of five keyword occurrences. There were 758 keywords detected, and 34 met the threshold specified above.

Figure 9 shows an overlay visualization of author keywords in six clusters with a total link of 482 and a total link strength of 1644. ‘Ergonomics’ in Cluster 2 is the keyword most frequently used by the author, with 147 co-occurrences linked to 33 other words and a total link strength of 278. The keyword in second place is ‘musculoskeletal disorders (MSDs)’, found in Cluster 2 with 124 co-occurrences, links with 32 other authors’ keywords, and a total link strength of 274. The remainder of the ten most used keywords by the authors are ‘dentists’ (88 co-occurrences with 31 links and a total link strength of 177), ‘dentistry’ (65 co-occurrences with 25 links and a total link strength of 113), ‘dental students’ (35 co-occurrences with 23 links and a total link strength of 78), ‘posture’ (28 co-occurrences with 20 links and a total link strength of 69), ‘risk factors’ (19 co-occurrences with 19 links and a total link strength of 54), ‘dental ergonomics’ (16 co-occurrences with 14 links and a total link strength of 22), ‘occupational hazards’ (16 co-occurrences with 19 links and a total link strength of 42), and ‘prevalence’ (16 co-occurrences with 15 links and a total link strength of 37).

Table 8 lists the authors’ keywords based on research subjects, research methods, occupational health and musculoskeletal disorders, ergonomics, and knowledge and education. The purpose of the grouping is to determine world research trends in the field of dental ergonomics [38].

#### 3.5.2. Analysis of the Co-Citation Network of Cited Authors

Co-citation represents two articles that are cited together by at least one article published later. In other words, if two articles are cited together by at least one article, then those two articles are called co-citations. The number of articles that cite the two articles mentioned together is called the frequency or strength of the co-citation [40]. These initial two articles have high co-citation power if more and more articles are published that cite these two articles. Co-citation patterns will change over time. Bibliographic coupling existed earlier than co-citation, and co-citation analysis is considered more recent in reflecting domain structure [41].

In this research, analysis of the co-citation network of authors cited using the VOSviewer application covered 14,317 authors. Setting the threshold of the minimum number of citations for an author as 20 citations, 164 authors were identified. For each of the 164 authors, the VOSviewer application calculated the total strength of co-citation relationships with other authors. Table 9 presents the 15 authors with the greatest co-citations and total link strengths.

## 4. Discussion

The Scopus database was used as the source of bibliographic data in this research because it has wide coverage, has good data quality and accuracy, provides various bibliometric analysis features, and is a data source that is verified and academically recognized [42]. Falagas et al. compared the strengths and weaknesses of PubMed, Scopus, Web of Science, and Google Scholar. PubMed and Google Scholar are free; while PubMed is optimal for biomedical research, Google Scholar’s accuracy is inconsistent. Scopus, meanwhile, has 20% greater citation analysis coverage compared to Web of Science [43]. Sing et al. compared three Web of Science databases, Scopus, and Dimensions. It was reported that almost all journals on the Web of Science can be found in Scopus and Dimensions. Meanwhile, Scopus indexes 66.07% more unique journals compared to Web of Science. Web of Science and Scopus coverage tends to be in the areas of life sciences, physical sciences, and technology, while Dimensions covers more social sciences and arts and humanities [44].

The results of the bibliometric analysis showed that up to April 2023, 682 articles about dentists and dental ergonomics indexed by Scopus could be identified. The number of articles per year varied greatly, where the most prominent decline in articles was in 1979, when only one article was published. More recently, articles on dental ergonomics have increased sharply, with 36 in 2020, 43 in 2021, and 40 by the end of 2022. In 2023, as of April, there were 10 articles.

This sharp increase stems from various studies providing scientific evidence of the high prevalence of MSDs in dentists and noting that these disorders have been found since the beginning of individual-focused dental studies [45,46,47]. These disorders are caused by awkward body postures, unergonomic instruments, poor environmental and system planning, and inadequate work practices. On the other hand, there is still little scientific evidence on the effectiveness of ergonomic educational interventions for improving body posture following induction as a dental student. This raises research questions as to why the prevalence of MSDs in dentists is so high and why ergonomic education interventions aimed at implementing healthy work postures have not had a significant impact [48].

The application of dental ergonomics is important because when working, dentists repeatedly assume sitting, standing, and static positions. Static postures are often used by dentists, such as bending the body forward, bending the neck forward, tilting towards the patient’s mouth, rotating the spine, and abducting the hands for a long time [49]. Static positions cause excessive contractions in several tissues, increasing muscle tension and thereby causing pain in the musculoskeletal system and peripheral nervous system [50]. In addition, the work involves high visual demands, which result in postural adaptations. In their work, dentists often assume a kyphotic posture, bending and turning the head to adjust their field of vision, with lumbar rotation and flexion. Therefore, the prevalence of MSDs in dentists is higher compared to that in other professions [51].

Other risk factors for MSDs include static and awkward neck and shoulder postures, repetitive movements with force in the hands and arms, poor lighting, the patient position not being appropriate to the dentist’s position, individual characteristics (physical condition, height, weight, general health, gender, age), and stress [52]. MSDs reduce an individual’s range of motion, grip strength, normal sensation, and even coordination of the musculoskeletal system [53]. MSDs in dentists begin with initial symptoms including pain, swelling, tenderness, numbness, and loss of strength [54]. In research in Saudi Arabia, neck and back pain were the main problems for dentists, which could start to be corrected in the process of dental education. So, it is important for dental schools to improve dental ergonomics training for their students [55].

The main goal of dental ergonomics is to reduce the risk of MSDs and to minimize the amount of physical and mental stress so that the quality of dentists’ work can be improved [56]. In addition, in the development of dental ergonomics research, the subjects are not only dentists and dental students but should also extend to dental hygienists [57], dental assistants [58], and dental technicians [59]. The progress of dental ergonomics cannot be separated from its history, where initially dentists worked in a standing position; however, since the 1960s, the four-handed dentistry system has been developed where dentists work in a sitting position [60]. Four-handed dentistry is a dental ergonomic effort to minimize unwanted movements and speed up dental treatment procedures [61].

Lietz et al., in their systematic review, considered various studies of ergonomic interventions to prevent MSDs in dental professionals. Of the 11 studies, 5 studies used ergonomic interventions in the form of using magnifying glasses or prismatic glasses; 2 studies used ergonomic dental chairs; 1 study used ergonomic dental instruments; and 3 studies provided ergonomic interventions in the form of ergonomic training. The results of all the included studies showed the important role of ergonomic interventions that can provide an improved work posture, increase work performance, and reduce the severity of MSDs in dental professionals [62].

Our assessment of the development of world trends in the field of dental ergonomics showed that the United States is ranked highest with 89 articles, followed by India (66 articles), Brazil (29 articles), Germany (25 articles), Saudi Arabia (22 articles), Sweden (22 articles), and the United Kingdom (22 articles). The United States is the country with the most articles written across countries and is the country with the highest number of citations. Nonetheless, the analysis of the most cited countries showed that not all countries with more articles have high citations. For instance, Greece is in the sixth position among the top 10 countries with the highest citations, with 287 citations obtained from its three articles, as can be seen in Table 3. Based on network visualization, five countries have cited articles from Greece, namely, Sweden (link strength: 10), Germany (link strength: 9), the United States (link strength: 7), Iran (link strength: 6), and India (link strength: 5), in order of link strength level.

The *Journal of The American Dental Association* (*JADA*) is the journal with the highest number of citations, and the next is *BMC Musculoskeletal Disorders*, receiving 460 and 421 citations, respectively. *JADA* is the leading open-access journal in the United States, which has been around since 1913 and has a Q2 ranking with an h-index of 128, with its subject areas specifically in dentistry and medicine. Since 2000, *BMC Musculoskeletal Disorders* has been an open-access journal, ranked Q2 with an h-index of 112 subject areas including orthopedics, sports medicine, and rheumatology [63]. In third place is *Work*, with a total of 338 citations. *Work* has been in existence since 1990 with subject areas covering prevention, assessment, and rehabilitation. It is an interdisciplinary journal ranked Q2 with an h-index of 58. In the fourth place is *Applied Ergonomics* which has 262 citations, meanwhile, with a Q1 ranking and an h-index of 119 indicated that this open-access journal is aimed explicitly at ergonomists and professionals who apply human factors in designing, planning, and managing technical and social systems. This bibliometric analysis showed that journal age, open-access status, topic, quality, and impact factors were essential in determining the number of document citations [64].

“The prevalence of musculoskeletal disorders in dentists” was the most frequently cited article and reached the top ranking with 274 citations. This was written by Alexopoulos E.C. in 2004, from the Department of Public Health, Technological Educational Institute of Athens, Greece and the Occupational Health Department, Hellenic Shipyards SA, Athens, Greece. The contents of this article are the results of a survey of 430 dentists in Thessaloniki, Greece, using the Nordic questionnaire to determine the occurrence of MSD complaints in the last 12 months, chronic MSD complaints for at least one month, MSD complaints that caused an inability to work, and whether the respondents sought medical treatment. The survey results showed that 62% of dentists experienced at least one MSD complaint, 30% of dentists experienced chronic MSD complaints, 16% stated they had been absent from work, and 32% of dentists sought medical treatment. From these results, it was concluded that dentists are at risk of experiencing MSDs related to the physical load of their work [65]. Ranked second was an article entitled ”Preventing musculoskeletal disorders in clinical dentistry: Strategies to address the mechanisms leading to musculoskeletal disorders” written by Valachi B., a physiotherapist who is one of the founders of Posturedontics, Portland, Oregon. In this article, strategies for preventing the development of MSDs in dentistry are presented, which aim to shape body posture and work ergonomically [66]. The third most cited article is “Reports of the body in dental student population” by Rising D.W., from the Department of Preventive and Restorative Dental Sciences, School of Dentistry, University of California, San Francisco. The research was conducted on 271 dental students in their fourth year, and the conclusion was that 70 percent of students had experienced MSD complaints since their third year as dental students [67].

In the author visualization of keywords up to 2023, the keyword “Human Factors and Ergonomics (HFE)” was not found, even though HFE has progressed very rapidly recently. HFE is a science that studies interactions among humans, tasks, and elements of work systems, with the aim of making humans better integrated in a system via adaptations to the environment for the individual. Dentists and other dental health professionals will function better in a more conducive environment. HFE interventions have the potential to improve the performance, health, and welfare of workers, including (1) physical ergonomics interventions (anthropometrics, anatomy, physiology, biomechanics); (2) organizational ergonomics intervention (organizational structure, policies, procedures); and (3) cognitive ergonomic interventions relating to mental processes (memory, reasoning, perception, motor reactions) [68,69]. The research into the application of HFE in the dental education system and dental practice is a future challenge for the world.

This bibliometric analysis research allowed us to produce various quantitative descriptive images of country citations, journals, articles, authors, and author keywords on the theme of ergonomics and dentistry. However, there are several limitations in bibliometric analysis. One is that open access to scientometric data is required. Access to data with sufficient accuracy is a fundamental limitation of bibliometric analysis. The important information required includes metadata, author data, affiliations, and citations. Another limitation is the possibility that the downloaded data are incomplete or duplicate data. The main obstacle in bibliometric analysis is the complexity and diversity of bibliographic data, meaning researchers need to be careful in understanding the various dimensions of the data. The number of citations is directly proportional to time, meaning that older papers tend to receive more citations than new papers.

## 5. Conclusions

Dental ergonomics research has developed rapidly to the stage where the world’s trending research topic is MSDs in dentists. This topic includes the prevalence of MSDs, along with their causes, factors related to causes, prevention, assessment, rehabilitation, evaluation, and ergonomic intervention. There has not yet been any research on ergonomic interventions that collaborate with human factors and ergonomics (HFE). The future research challenge is to apply HFE science to improve the health, safety, efficiency, and quality of dentists’ work.

## Figures and Tables

**Figure 1 ijerph-21-00493-f001:**
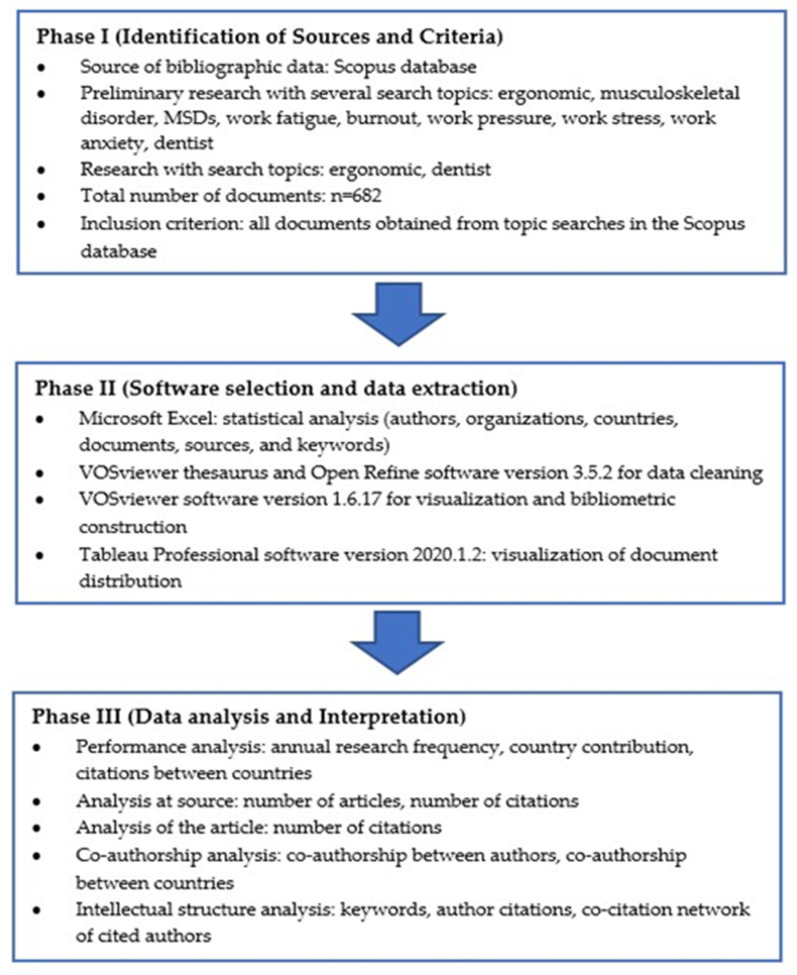
Research methods and flow.

**Figure 2 ijerph-21-00493-f002:**
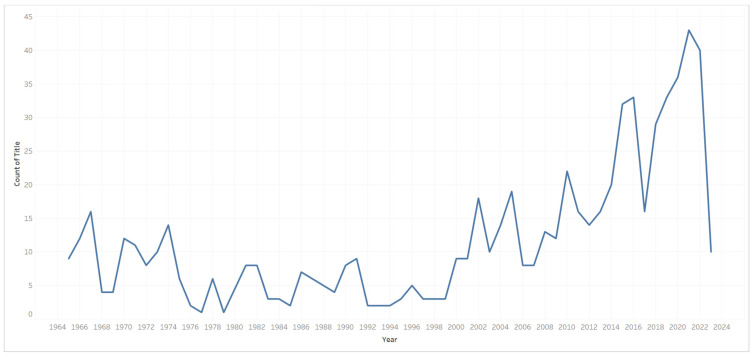
The trend of a count of titles for the year. The year and count of titles filter the view. The year ranges from 1965 to 2023. The count of the title filter ranges from 1 to 43.

**Figure 3 ijerph-21-00493-f003:**
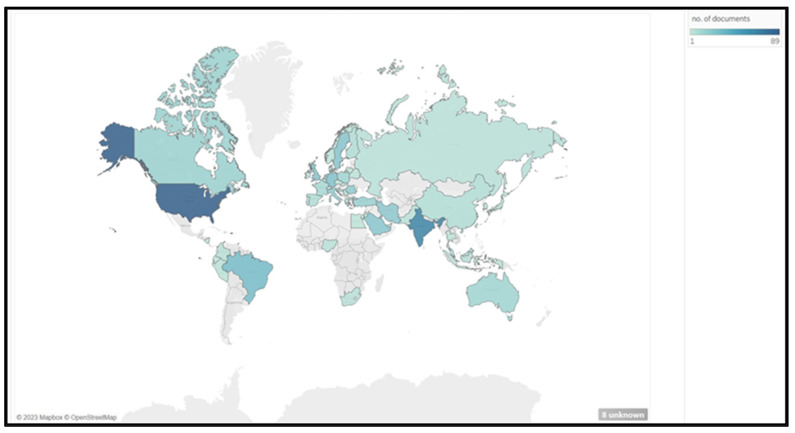
Distribution of documents by country.

**Figure 4 ijerph-21-00493-f004:**
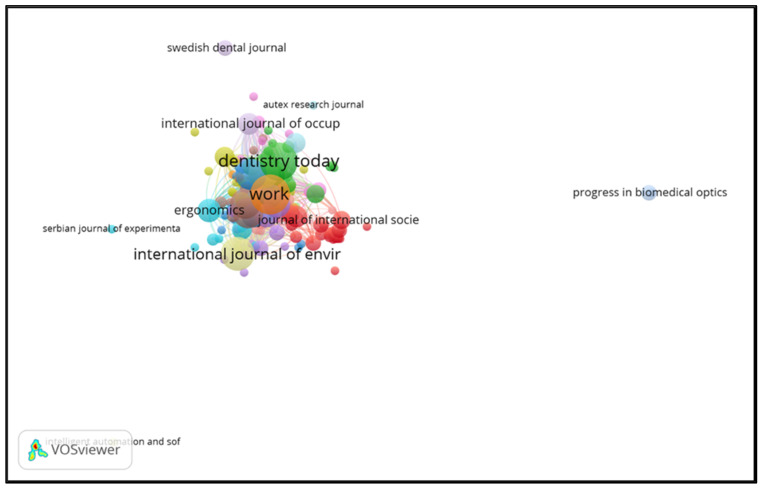
Bibliometric analysis of sources.

**Figure 5 ijerph-21-00493-f005:**
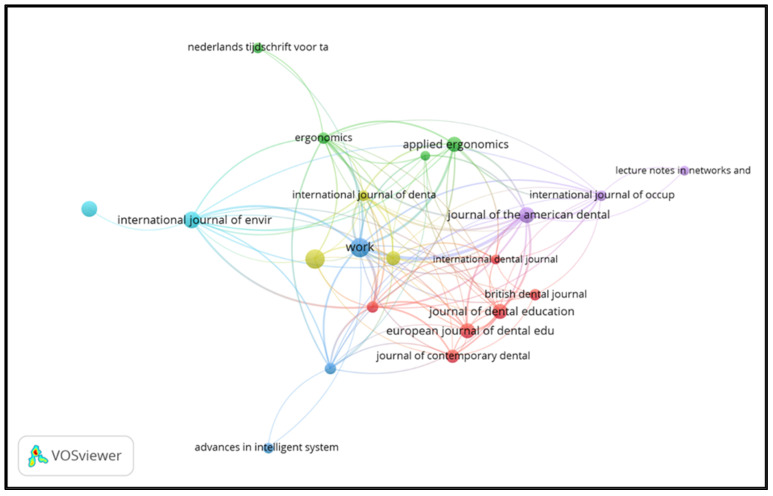
Bibliometric analysis of the number of source citations.

**Figure 6 ijerph-21-00493-f006:**
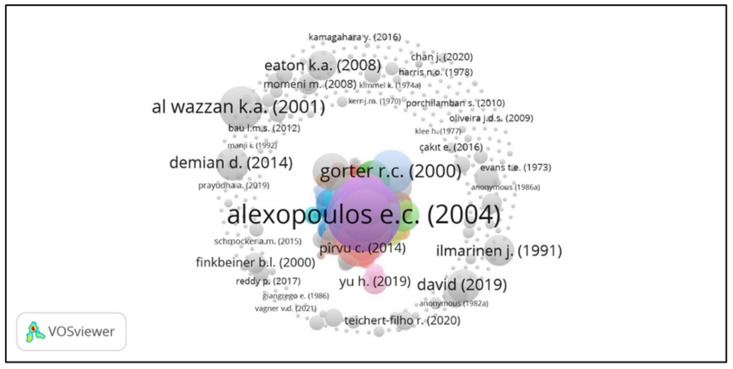
Network visualization in the analysis of the most frequently cited documents. There are two groups, namely, the first group includes colorful circles, dominated by Alexopoulos E.C. (2004) with the largest circle, and the second group includes transparent gray circles, dominated by Al Wazzan K.A. (2001) with the largest circle. This coloring difference indicates no relationship between the first and second groups. Regarding circle size, the higher the citation weight, the bigger the circle [35].

**Figure 7 ijerph-21-00493-f007:**
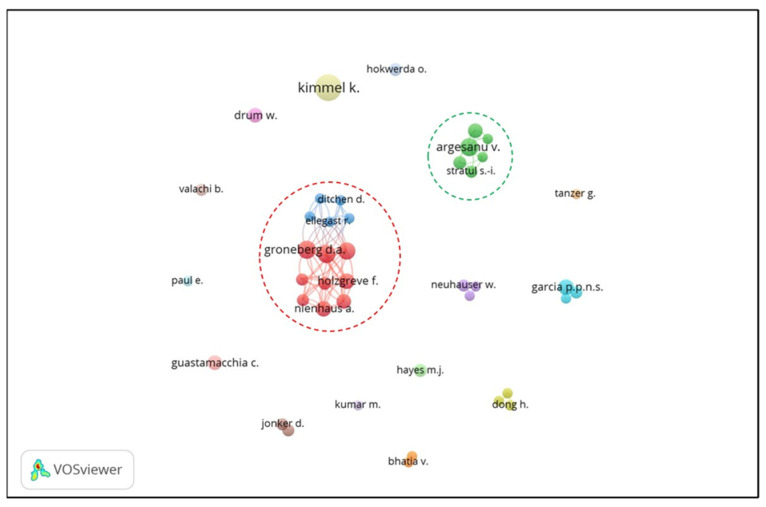
Co-authorship network visualization: in the red circle is the combined co-authorship of Clusters 1 and 3; meanwhile, in the green circle, the co-authorship of Cluster 2 is shown.

**Figure 8 ijerph-21-00493-f008:**
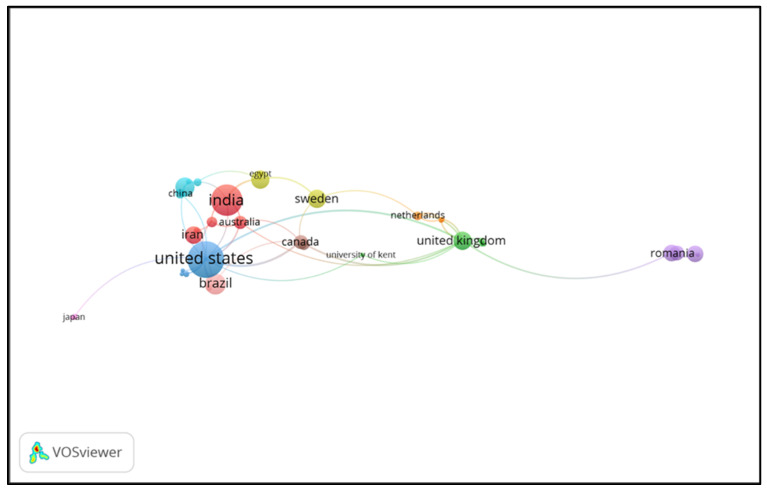
Bibliometric analysis of co-authorship by country.

**Figure 9 ijerph-21-00493-f009:**
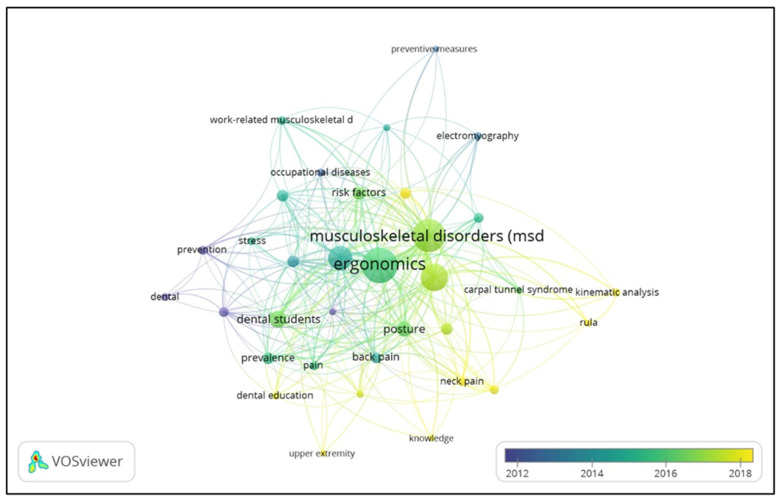
Overlay visualization map of author keywords from the 1990s to 2023. The color of the circles ranges from blue, which indexes articles with publication years around 2012, to green for 2014–2016, to yellow for 2018, to the latest year, 2023 [39].

**Table 1 ijerph-21-00493-t001:** Data mining with several topics in preliminary research.

Boolean Search Sentences	Number of Documents
(TITLE-ABS-KEY (ergonom*) AND TITLE-ABS-KEY (dentist*))	634
(TITLE-ABS-KEY (“musculoskeletal disorder*” OR ”MSDs*”) AND TITLE-ABS-KEY (dentist*))	372
(TITLE-ABS-KEY (“work fatigue*” OR ”burnout*”) AND TITLE-ABS-KEY (dentist*))	317
(TITLE-ABS-KEY (“work pressure*” OR ”work stress*” OR “work anxiety*”) AND TITLE-ABS-KEY (dentist*))	43

**Table 2 ijerph-21-00493-t002:** The 20 countries with the most significant number of documents.

Rank	Country	Region	Cluster	Docs.	Citations	Links	Total Link Strength
1	United States	America	12	89	1321	53	9801
2	India	Asia	3	66	577	51	10,642
3	Brazil	America	4	29	223	49	4335
4	Germany	Europe	7	25	227	50	7187
5	Saudi Arabia	Asia	4	22	288	51	4634
6	Sweden	Europe	4	22	633	49	3407
7	United Kingdom	Europe	5	22	252	53	2742
8	Iran	Asia	4	21	307	49	6730
9	Romania	Europe	9	19	97	44	803
10	Italy	Europe	9	18	169	49	2301
11	Turkey	Eurasia	3	15	152	48	2776
12	Canada	America	8	14	209	41	715
13	Poland	Europe	3	13	206	47	1446
14	Australia	Oceania	8	12	236	49	5514
15	France	Europe	8	9	29	48	1218
16	Malaysia	Asia	10	8	120	49	4210
17	Spain	Europe	1	6	109	48	1479
18	Finland	Europe	6	6	107	44	358
19	China	Asia	7	6	41	41	623
20	Croatia	Europe	2	6	25	33	180

**Table 3 ijerph-21-00493-t003:** The 20 countries with the highest number of citations.

Rank	Country	Region	Cluster	Docs.	Citations	Links	Total Link Strength
1	United States	America	1	89	1321	43	381
2	Sweden	Europe	2	22	633	37	197
3	India	Asia	5	66	577	36	266
4	Iran	Asia	4	21	307	30	160
5	Saudi Arabia	Asia	1	22	288	30	108
6	Greece	Europe	2	3	287	30	78
7	United Kingdom	Europe	2	22	252	24	71
8	Australia	Oceania	6	12	236	34	147
9	Germany	Europe	8	25	227	30	139
10	Brazil	America	7	29	223	30	106
11	Canada	America	4	14	209	22	69
12	Poland	Europe	3	13	206	19	51
13	Netherlands	Europe	4	5	178	18	35
14	Italy	Europe	10	18	169	27	80
15	Turkey	Eurasia	5	15	152	20	70
16	Malaysia	Asia	3	8	120	24	90
17	Spain	Europe	2	6	109	21	52
18	Finland	Europe	9	6	107	18	28
19	Romania	Europe	12	19	97	9	19
20	South Korea	Asia	2	3	79	21	45

**Table 4 ijerph-21-00493-t004:** Top-ranking sources with a minimum of 5 documents.

Rank	Sources	Country	ISSN	Docs.	Citations	h-Index	SJR (2022)	Q	Publication Type
1	*Work*	Netherlands	18759270 10519815	19	338	58	0.509	Q2	Journals
2	*Dentistry Today*	United States	87502186	18	73	27	0.102	Q4	Journals
3	*International Journal of Environmental Research and Public Health*	Switzerland	16617827 16604601	14	80	167	0.828	Q2	Journals
4	*Journal of The American Dental Association*	United States	00028177 19434723	12	460	128	0.520	Q2	Journals
5	*Applied Ergonomics*	United Kingdom	18729126 00036870	11	262	119	0.922	Q1	Journals
6	*Journal Of Dental Education*	United States	00220337 19307837	11	126	76	0.558	Q2	Journals
7	*European Journal of Dental Education*	United Kingdom	16000579 13965883	11	115	49	0.523	Q2	Journals
8	*Journal Of Contemporary Dental Practice*	United States	15263711	9	194	47	0.295	Q3	Journals
9	*BMC Musculoskeletal Disorders*	United Kingdom	14712474	7	421	112	0.716	Q2	Journals
10	*Ergonomics*	United Kingdom	00140139 13665847	7	98	124	0.676	Q1	Journals
11	*British Dental Journal*	United Kingdom	00070610 14765373	7	50	91	0.506	Q2	Journals
12	*Indian Journal of Dental Research*	India	19983603 09709290	7	150	50	0.264	Q3	Journals
13	*International Journal of Dental Hygiene*	United Kingdom	16015029 16015037	6	164	44	0.635	Q1	Journals
14	*International Journal of Occupational Safety and Ergonomics*	United Kingdom	10803548	6	80	43	0.513	Q2	Journals
15	*Advances In Intelligent Systems and Computing*	Germany	21945365 21945357	6	16	58	Discontinued (2021)	-	Book Series
16	*Annals of DAAAM and Proceedings of The International DAAAM Symposium*	Austria	17269679	6	3	19	Not yet assigned a quartile	-	Conferences and Proceedings
17	*International Dental Journal*	Netherlands	00206539 1875595X	5	63	73	0.733	Q1	Journals
18	*Journal of Clinical and Diagnostic Research*	India	0973709X 2249782X	5	52	64	Discontinued (2018)	-	Journals
19	*Lecture Notes in Networks and Systems*	Switzerland	23673370 23673389	5	1	27	0.151	Q4	Book Series

**Table 5 ijerph-21-00493-t005:** Top 15 ranked sources with the most citations.

Rank	Sources	Country	ISSN	Citations	Docs	h-Index	SJR (2022)	Q	Subject Area
1	*Journal of The American Dental Association*	United States	00028177 19434723	460	12	128	0.520	Q2	Dentistry Medicine
2	*BMC Musculoskeletal Disorders*	United Kingdom	14712474	421	7	112	0.716	Q2	Medicine
3	*Work*	Netherlands	18759270 10519815	338	19	58	0.509	Q2	Medicine
4	*Applied Ergonomics*	United Kingdom	18729126 00036870	262	11	119	0.922	Q1	Engineering Health Professions Social Sciences
5	*Journal of Contemporary Dental Practice*	United States	15263711	194	9	47	0.295	Q3	Dentistry
6	*Journal of The California Dental Association*	United States	10432256	188	10	44	Not yet assigned a quartile	-	Dentistry Medicine
7	*International Journal of Dental Hygiene*	United Kingdom	16015029 16015037	164	6	44	0.635	Q1	Dentistry
8	*Indian Journal of Dental Research*	India	19983603 09709290	150	7	50	0.264	Q3	Dentistry Medicine
9	*Swedish Dental Journal*	Sweden	03479994	134	3	37	Not yet assigned a quartile	-	Dentistry Medicine
10	*Journal of Dental Education*	United States	00220337 19307837	126	11	76	0.558	Q2	Dentistry Medicine Social Sciences
11	*European Journal of Dental Education*	United Kingdom	16000579 13965883	115	11	49	0.523	Q2	Dentistry Social Sciences
12	*Indian Journal of Public Health Research and Development*	India	09765506 09760245	111	4	21	Not yet assigned a quartile	-	Medicine
13	*Annals of Agriculture and Environmental Medicine*	Poland	12321966 18982263	107	1	61	0.389	Q3	Agricultural and Biological Sciences Environmental Science Medicine
14	*Medicina Oral Patologia Oral y Cirugia Bucal*	Spain	16986946 16984447	106	2	66	0.587	Q2	Dentistry Medicine
15	*Journal of Occupational Health*	Japan	13489585 13419145	103	3	67	0.689	Q2	Medicine

**Table 6 ijerph-21-00493-t006:** Top 15 most cited documents.

Rank	Title	Authors	Journal	Year	Citations
1	Prevalence of musculoskeletal disorders in dentists	Alexopoulos E.C.; Stathi I.C.; Charizani F.	*BMC Musculoskeletal Disorders*	2004	274
2	Preventing musculoskeletal disorders in clinical dentistry: Strategies to address the mechanisms leading to musculoskeletal disorders	Valachi B.; Valachi K.	*Journal of The American Dental Association*	2003	139
3	Reports of body pain in dental student population	Rising D.W.; Bennett B.C.; Hursh K.; Plesh O.	*Journal of The American Dental Association*	2005	114
4	Disorders of the musculoskeletal system among dentists from the aspects of ergonomics and prophylaxis	Szymańska J.	*Annals of Agriculture and Environmental Medicine*	2002	107
5	Musculoskeletal disorders of the neck and shoulder in the dental professions	Morse T.; Bruneau H.; Dussetschleger J.	*Work*	2010	102
6	Work characteristics and upper extremity disorders in female dental health workers	Lindfors P.; von Thiele U.; Lundberg U.	*Journal of Occupational Health*	2006	99
7	Burnout and health among Dutch dentists	Gorter R.C.; Eijkman M.A.; Hoogstraten J.	*European Journal of Oral Sciences*	2000	99
8	Back & neck problems among dentists and dental auxiliaries	Al Wazzan K.A.; Almas K.; Al Shethri S.E.; Al-Qahtani M.Q.	*Journal of Contemporary Dental Practice*	2001	93
9	Work-related musculoskeletal disorders among dentists-a questionnaire survey	Kierklo A.; Kobus A.; Jaworska M.; Botuliński B.	*Annals of Agriculture and Environmental Medicine*	2011	90
10	The effect of tool handle shape on hand muscle load and pinch force in a simulated dental scaling task	Dong H.; Loomer P.; Barr A.; Laroche C.; Young E.; Rempel D.	*Applied Ergonomics*	2007	82
11	Low back problems and possible improvements in nursing jobs	Vieira E.R.; Kumar S.; Coury H.J.C.G.; Narayan Y.	*Journal Of Advanced Nursing*	2006	70
12	Evaluating dental office ergonomic risk factors and hazards	Bramson J.B.; Smith S.; Romagnoli G.	*Journal of The American Dental Association*	1998	61
13	Evaluation of ergonomic interventions to reduce musculoskeletal disorders of dentists in the Netherlands	Droeze E.H.; Jonsson H.	*Work*	2005	61
14	Perceived musculoskeletal symptoms among dental students in the clinic work environment	Thornton L.J.; Barr A.E.; Stuart-Buttle C.; Gaughan J.P.; Wilson E.R.; Jackson A.D.; Wyszynski T.C.; Smarkola C.	*Ergonomics*	2008	59
15	Pain and discomfort in the musculoskeletal system among dentists. A prospective study	Rundcrantz B.L.; Johnsson B.; Moritz U.	*Swedish Dental Journal*	1991	58

**Table 7 ijerph-21-00493-t007:** Top 10 countries in co-authorship.

Rank	Country	Cluster	Docs.	Citations	Links	Total Link Strength	Collaborating Countries
1	United States	3	89	1321	15	25	Brazil, Nigeria, Iran, Malaysia, Australia, India, Germany, China, Canada, Japan, United Kingdom
2	India	1	66	577	5	9	Malaysia, Australia, United Arab Emirates, Saudi Arabia, United States
3	Brazil	10	29	223	3	7	United States, Canada, Portugal
4	Germany	6	25	227	4	4	United States, United Arab Emirates, Lithuania, China
5	United Kingdom	2	22	252	11	19	Netherlands, South Africa, Belgium, Trinidad and Tobago, Canada, Australia, United States, Romania
6	Saudi Arabia	4	22	288	5	8	United Arab Emirates, Sweden, India, Egypt
7	Sweden	4	22	633	3	4	Saudi Arabia, Canada, Netherlands
8	Iran	1	21	307	2	3	United States, South Korea
9	Romania	5	19	97	3	4	United Kingdom, Italy, Turkey
10	Italy	5	18	169	2	2	Romania

**Table 8 ijerph-21-00493-t008:** Grouping of author keyword visualization overlays up to 2023.

Category	Authors’ Keywords	Co-Occurrences	Link	Total Link Strength	Avg. Pub. Year.
Research subjects	Dentists	88	31	177	2017.07
	Dental students	35	23	78	2016.40
	Dental hygienists	12	12	29	2015.25
	Dental staff	6	16	24	2009.17
Method	Survey and questionnaires	12	18	30	2010.75
	Electromyography	8	7	16	2013.25
	Kinematic analysis	8	5	16	2019.50
	Rula	7	7	16	2018.43
Occupational health and MSDs	Musculoskeletal disorders (MSDs)	124	32	274	2016.85
	Risk factors	19	19	54	2016.42
	Occupational hazards	16	19	42	2013.81
	Prevalence	16	15	37	2015.44
	Occupational health	15	15	39	2014.80
	Back pain	13	15	34	2014.46
	Neck pain	10	13	32	2018
	Pain	10	14	28	2015.50
	Prevention	10	12	21	2011.80
	Work-related musculoskeletal disorders	10	9	21	2015
	Lower back pain	9	11	28	2017.78
	Occupational diseases	8	10	19	2012.75
	Stress	8	11	15	2014.88
	Cumulative trauma disorders	7	10	16	2014.57
	Musculoskeletal system	6	17	22	2017.50
	Carpal tunnel syndrome	6	9	21	2016.17
	Upper extremity	5	5	7	2018
Ergonomics	Ergonomics	147	33	278	2015.80
	Dentistry	65	25	113	2014.57
	Posture	28	20	69	2016.25
	Dental ergonomics	16	14	22	2017.19
	Magnification	14	11	32	2018.79
	Dental	8	5	7	2010
	Preventive measurements	5	5	8	2013
Knowledge and Education	Dental education	6	10	16	2017.67
	Knowledge	5	6	11	2020.60

**Table 9 ijerph-21-00493-t009:** Top 15 co-cited authors in ergonomic and dentist references.

Rank	Authors	Cluster	Co-Citations	Links	Total Link Strength
1	Smith, D.R.	4	244	163	9793
2	Valachi, B.	2	167	163	5822
3	Moritz, U.	1	143	163	5093
4	Valachi, K.	2	126	163	4576
5	Cockrell, D.	4	121	163	4606
6	Hayes, M.J.	4	121	163	4901
7	Leggat, P.A.	2	115	163	4940
8	Johnsson, B.	1	110	162	3828
9	Akesson, I.	1	104	163	4013
10	Skerfving, S.	1	92	160	3565
11	Finsen, L.	1	89	163	3229
12	Christensen, H.	1	84	163	3065
13	Ohlendorf, D.	3	76	155	5471
14	Szymanska, J.	2	75	158	2135
15	Kedjarune, U.	2	73	163	3160

## Data Availability

Research data can be obtained via the corresponding author’s email.

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
