# Peer review of "World Trends in Dental Ergonomics Research: A Bibliometric Analysis"

_ijerph, 2024, doi:10.3390/ijerph21040493_

Round 1

Reviewer 1 Report

Comments and Suggestions for Authors

Dear Authors,

The work is a very good attempt in this field; however English language editing has to be done for better readability. Additional comments are in the PDF file. Modifying the manuscript will make it more useful and appropriate.

Comments on the Quality of English Language

English language editing is required for better clarity. 

Author Response

Dear Reviewer 1

This is my revision based on your review. I marked this revision with a green highlighter.

Thank you for your attention.

  1. Reviewer comment:

Page: 12; Row: 250

VOSviewer: Can elaborate about VOSviewer to the readers so that the topic can be related better by the readers

Author's response and answer:

Page: 2; Rows: 80-89

In this bibliometric analysis, VOSviewer software is used to create a publication network data map accompanied by data visualization that includes co-authorship, co-occurrence, citation, bibliographic coupling, or co-citation links. The use of this software helps researchers, librarians, and publication database managers to obtain a network map of scientific publications including authors or researchers, organizations, countries, and keywords [27-29]. VOSviewer users can import bibliographic databases from Scopus, PubMed, or Web of Science [30]. Bibliometric analysis techniques have developed over time and continuously, to measure the impact of publishing articles within the scientific community. All data are presented in the form of mapping to describe the relationships between nodes in the expanded analysis [31].

  1. Reviewer comment:

Page: 12-13; Rows: 257-260

In contrast to that year, since 2020, publications on dental ergonomics have increased sharply by 36 publications; in 2021, there will be 43 publications, and by the end of 2022, there will be 40 publications. In 2023, up to April, there have been ten publications.

-     Any particular analysis / thoughts on why the sharp rise of such publications? Any analysis?

-     Change the English language

Author's response and answer:

Page 16; Rows: 329-337

  • This sharp increase is related to various studies providing scientific evidence that confirmed the high prevalence of MSDs in dentists and that these disorders have even been found since the beginning of individuals’ dental studies [45-47]. These disorders are caused by awkward body postures, unergonomic instruments, poor environmental and system planning, and inadequate work practices. On the other hand, there is still little scientific evidence that reports the effectiveness of ergonomic educational interventions on body posture following induction as a dental student. This raises research questions as to why the prevalence of MSDs in dentists is so high and why ergonomic education inter-ventions to implement healthy work postures have not had a significant impact [48].
  • This manuscript has undergone English language editing by MDPI. The text has been checked for correct use of grammar and common technical terms, and edited to a level suitable for reporting research in a scholarly journal. MDPI uses experienced, native English speaking editors. Full details of the editing service can be found at

â–º https://www.mdpi.com/authors/english.

  1. Reviewer comment:

Page: 13; Rows: 299-300

Prevalence of musculoskeletal disorders in dentists' was the most frequently cited article and reached the top ranking with 274 citations:

  • Can add a note on various guidelines proposed for preventing MSD from various studies

Author's response and answer:

Page: 16; Rows: 367-374

Lietz et al., in their systematic review, reviewed various studies of ergonomic interventions to prevent MSDs in dental professionals. Of the 11 studies, 5 studies used ergonomic interventions in the form of using magnifying glasses or prismatic glasses; 2 studies used ergonomic dental chairs; 1 study used ergonomic dental instruments; and 3 studies provided ergonomic interventions in the form of ergonomic training. The results of all the included studies show the important role of ergonomic interventions that can provide improved work posture, increase work performance, and reduce the severity of MSDs in dental professionals [62]

  1. Reviewer comment:

Page: 14; Rows: 321-328

The application of dental ergonomics is important because dentists repeatedly assume sitting, standing, and static positions when working. Dentists often use static postures such as bending the body forward, bending the neck forward and tilting towards the patient's mouth, twisting the spine, and abducting the hands for a long time [44]. Static positions cause excessive contraction of several tissues, increasing muscle tension and causing musculoskeletal and peripheral nervous pain [45]. High visual demands result in postural adaptations. Dentists often assume a kyphotic posture, bending and turning the head to adjust their field of vision with lumbar rotation and flexion. Therefore, the prevalence of MSDs in dentists is higher compared to other professions [46].

  • Can be shifted at the beginning of the discussion

Author's response and answer:

Thank you for the suggestion and it has been implemented, on page 16, rows: 338-347

  1. Reviewer comment:

Page: 14; Rows: 330-336

Other risk factors for MSDs include static and awkward neck and shoulder postures, repetitive movements with force in the hands and arms, poor lighting, patient position not appropriate to the dentist's position, individual characteristics (physical condition, height, weight, general health, gender, age) and stress [47]. MSDs will reduce the range of

motion grip strength, eliminate normal sensation, and even coordination of the musculoskeletal system [48]. MSDs in dentists begin with initial symptoms including pain, swelling, tenderness, numbness, and loss of strength [49]. In research in Saudi Arabia, neck and back pain were the main problems for dentists, which could start with dental education.

  • Risk factor analysis can be better placed in the beginning of the dicussion

Author's response and answer:

Thank you for the suggestion and it has been implemented, on page 16, rows: 348-357

Reviewer 2 Report

Comments and Suggestions for Authors

The manuscript proposes an interesting study on the state of the art or research addressing ergonomics among dentists by means of bibliometric analysis.

The manuscript is well-organized and the subject fits with the journal topics providing valuable research insights.

However, before considering it for publication several improvements are needed.

First, the abstract should be more descriptive by illustrating in a qualitative manner what the research is about, which methods are used, and which are the main findings.

Then, in the introduction, research motivations should be elaborated more in order to bring to light the novelty of the study compared to the extant literature.

Also the materials and methods section has to be expanded by describing more in detail the review criteria used in the study and the tools for the bibliometric analysis. Figure 1 is unclear and contains some typos: it should be improved.

The analysis of results is quite clear and some additional figures/diagrams could augment its quality.

In the discussion of results practical findings of the study need to be elaborated more, as well as the study limitations.

Some additional comments are attached.

Comments on the Quality of English Language

In the text some typos can be found. Hence, language proofreading is needed.

Author Response

Dear Reviewer 2

This is my revision based on your review. I marked this revision with a yellow highlighter.

Thank you for your attention.

  1. Reviewer comment:

The manuscript proposes an interesting study on the state of the art of research addressing ergonomics among dentists by means of bibliometric analysis.

Author's response and answer:

Thank you for the review

  1. Reviewer comment:

The manuscript is well-organized and the subject fits with the journal topics providing valuable research insights.

Author's response and answer:

Thank you for the review

  1. Reviewer comment:

However, before considering it for publication several improvements are needed.

Author's response and answer:

Thank you for the review

  1. Reviewer comment:

First, the abstract should be more descriptive by illustrating in a qualitative manner what the research is about, which methods are used, and which are the main findings.

Author's response and answer:

Page: 1; Row: 9-25

Abstract: Background: Dental ergonomics provides an overview of dentists' work efficiency. Objective: The objective of this study was to obtain quantitative information and a visualization of the network of scientific publications on the topic of ergonomics and dentistry using bibliometric analysis.  Methods: Data mining was conducted using the Scopus database and Boolean expressions (ergonom* AND dentist*) on April 14 2023. Data extraction and analysis were performed using Open Refine version 3.5.2., VOSviewer version 1.6.17., VOSviewer thesaurus, Microsoft Excel, and Tableau Professional version 2020.1.2. Results: A total of 682 documents were identified, with the United States having the largest number of documents and citations (89 documents, 1321 citations). Work, Dentistry Today, and the International Journal of Environmental Research and Public Health, were the top three sources. Ergonomics and musculoskeletal disorders (MSDs) are two of the very prominent keywords, with research topics covering prevalence, causes, factors related to causes, prevention, assessment, rehabilitation, evaluation, and intervention. There was no research on ergonomic interventions that collaborated with human factors and ergonomics (HFE). Conclusion: The trend in dental ergonomics research topics around the world is centered on MSDs. The future research challenge is to apply HFE science to improve the health, safety, efficiency, and quality of dentists' work.

  1. Reviewer comment:

Then, in the introduction, research motivations should be elaborated more in order to bring to light  the novelty of the study compared to the extant literature. The justification of the study and the references provided to support the study are not sufficient. Moreover, the Authors focus on the problem of work-related musculoskeletal disorders (WMSDs), but this issue is not specifically addressed in the review.

Author's response and answer:

Page: 1

  • The dental profession involves repetitive movements combined with forceful movements, awkward postures, and inadequate recovery time. This profession requires precision and high visual requirements due to the narrow working area, i.e., the oral cavity. Therefore, dentists are at high risk of developing work-related musculoskeletal disorders (WMSDs) [2].
  • WMSDs are a significant problem for dentists and have been discovered early in their careers, even among dental students.

Page: 2, Row: 52-80

The most common MSDs in dentists are back pain, followed by neck pain, shoulder pain, high tension of the trapezius muscle, tendinitis, carpal tunnel syndrome, pinched nerves, early arthrosis, myopia, and auditive changes [8,9]. Pain in the muscles is an alarm in the body before the risk of paralysis and injury occurs, which has the potential to end a career early [10]. In their systematic review, Bret and Gorce reported that the highest prevalence of MSDs in dentists was in the lower back (>60%), shoulders, and upper extremities (35-55%). The main cause was an awkward posture repeated over a long time [11].  Soo et al. reported that dentists' susceptibility to MSDs reached 68% to 100% in various parts of the body, especially in the neck (26%-92%), shoulders (25%-92, 7%), and lower back (29% to 94.6%). The causes and problems of MSDs are multifactorial; several risk factors occur for female dentists (57.1%), with awkward posture (50%), with long periods of work (30%), and for specialist dentists (42.9%) [12]. To maximize the performance of dentists, human factors and ergonomics (HFE) is a special study area aimed at improving the health, safety, efficiency, and quality of dentists' work while also having a positive impact on patient safety. HFE interventions in health services are categorized into the following: (1) physical ergonomic interventions, (2) cognitive ergonomic interventions, and (3) organizational ergonomic interventions [13].

Scientific publications on ergonomics related to dentists have been around since the 1960s and have consistently increased from year to year. The research subjects are not only dentists but also dental assistants [14], dental hygienists [15], and dental students [16].  The scope of the research is very broad, including the work environment [17,18] and ergonomic interventions [16,19–22]. These research articles have contributed to producing dental ergonomic principles so that dentists always work with an “ergonomic culture” [23] and provide strategies for preventing MSDs [24,25].

Based on this background, a bibliometric analysis was carried out on the topic of ergonomics and dentists. Bibliometric analysis is a statistical tool for mapping the highest and current levels of scientific development and identifying research gaps and trends for various purposes, such as searching for research opportunities and supporting scientific research [26]. 

Page: 2, Row: 90-92

The aim of this research is to obtain quantitative information and visual information on world trends in ergonomics research related to dentists or dental ergonomics research in all Scopus indexed publications up to 2023.

  1. Reviewer comment:

Also the materials and methods section has to be expanded by describing more in detail the review  criteria used in the study and the tools for the bibliometric analysis. The Authors have to justify the proposed review procedure in a scientifically sound manner. Figure 1 is unclear and contains some typos: it should be improved. The flow process of the activities characterizing the review is too general. A clear time frame has to be defined and justified: from 1965 to April 2023 appears quite unusual considering the technology and knowledge development of recent decades.

Author's response and answer:

Page: 3, Row: 98-126

This research was carried out in two stages. The first stage was an exploratory stage of searching for research topics using several keywords with Boolean expressions in the Scopus database. The purpose of this preliminary research was to find research topics with keywords that can provide the most data information. A preliminary research topic search was carried out on 5 April 2022 using Boolean sentences in the Scopus electronic database. To search for phrases in Scopus double quotes are used ("), wildcards (*), and Boolean operators (OR, AND, NOT). The purpose of double quotes is to tell Scopus that these are “loose phrases" meaning that the words must be together. The use of wildcards (*) to represent a number of characters and Boolean operators are used to expand or narrow search parameters when using databases or search engine. The default search field in Scopus uses TITLE-ABS-KEY because the Scopus database is an abstract indexer only [32,33].          

Table 1. Data mining with several topics in preliminary research

Boolean Search Sentences

Number of documents

(TITLE-ABS-KEY (ergonom*) AND TITLE-ABS-KEY (dentist*))

634

(TITLE-ABS-KEY (“musculoskeletal disorder*” OR ”MSDs*”) AND TITLE-ABS-KEY (dentist*))

372

(TITLE-ABS-KEY (“work fatigue*” OR ”burnout*”) AND TITLE-ABS-KEY (dentist*))

317

(TITLE-ABS-KEY (“work pressure*” OR ”work stress*” OR “work anxiety*”) AND TITLE-ABS-KEY (dentist*))

43

Based on the results of data mining with several topics in preliminary research, the topics of ergonomics and dentistry were selected which provided the largest number of documents. During this research, data mining was carried out on the Scopus database using Boolean expressions (ergonom* AND dentist*) on 14 April 2023. The search results were exported into a Comma Separated Value (CSV) file in Microsoft Excel [34]. Microsoft Excel software was also used to analyze all information from recruited sources. To work on visualization and bibliometric construction, VOSviewer software version 1.6.17 was used and to clean the data, Thesaurus_text in VOSviewer and Open Refine software version 3.5.2 were used.

In the thesaurus step, keywords that have the same meaning (synonyms/hyponyms) were combined or deleted. The document distribution visualization tool was Tableau Professional software version 2020.1.2. The bibliography analysis attributes in VOSviewer software include co-authorship, co-occurrence, citation, bibliography coupling, co-citation of authors, organizations, and countries. The bibliometric analysis flow can be seen in Figure 1.

Figure 1. (The revised image can be seen in the manuscript on page 4)

  1. Reviewer comment:

The analysis of results is quite clear but the inclusion of some additional figures/diagrams could augment its quality.

 Author's response and answer:

Additional images are as follows:

Page 5; Row: 147-149 --Figure 3. Distribution of documents by country.

Page 7: Row: 180-181 --Figure 4. Bibliometric analysis of sources.

Page 7; Row: 182-184 --Figure 5. Bibliometric analysis of the number of source citations.

Page 12; Row: 253-255 -- Figure 8. Bibliometric analysis of co-authorship by country.

  1. Reviewer comment:

In the discussion of results, the practical findings of the study need to be elaborated more, as well as the study's limitations. Apart from statistical outcomes, the Authors should provide more information about research trends concerning the improvement of ergonomics in the dental profession.

Some additional comments are attached.

Author's response and answer:

Page 16; Row: 322-374

 The results of the bibliometric analysis show that up to April 2023, 682 articles about dentists and dental ergonomics indexed by Scopus were identified. The number of articles per year varies greatly, where the most prominent decline in articles was in 1979, when only one article was published. In contrast to that year, since 2020, publications on dental ergonomics have increased sharply by 36 publications; in 2021, there will be 43 publications, and by the end of 2022, there will be 40 publications. In 2023, up to April, there were 10 publications.

This sharp increase is related to various studies providing scientific evidence that confirmed the high prevalence of MSDs in dentists and that these disorders have even been found since the beginning of individuals’ dental studies [45-47]. These disorders are caused by awkward body postures, unergonomic instruments, poor environmental and system planning, and inadequate work practices. On the other hand, there is still little scientific evidence that reports the effectiveness of ergonomic educational interventions on body posture following induction as a dental student. This raises research questions as to why the prevalence of MSDs in dentists is so high and why ergonomic education interventions to implement healthy work postures have not had a significant impact [48].

The application of dental ergonomics is important because when working, dentists repeatedly assume sitting, standing, and static positions. Static postures are often used by dentists, such as bending the body forward, bending the neck forward, and tilting towards the patient's mouth, rotating the spine and abducting the hands for a long time [49]. Static positions cause excessive contractions in several tissues, increasing muscle tension and thereby causing pain in the musculoskeletal system and peripheral nervous system [50]. In addition, the work involves high visual demands, which result in postural adaptations. In their work, dentists often assume a kyphotic posture, bending and turning the head to adjust their field of vision, with lumbar rotation and flexion. Therefore, the prevalence of MSDs in dentists is higher compared to that in other professions [51].

Other risk factors for MSDs include static and awkward neck and shoulder postures, repetitive movements with force in the hands and arms, poor lighting, the patient position not being appropriate to the dentist's position, individual characteristics (physical condition, height, weight, general health, gender, age), and stress [52]. MSDs reduce an individual’s range of motion, grip strength, normal sensation, and even coordination of the musculoskeletal system [53]. MSDs in dentists begin with initial symptoms including pain, swelling, tenderness, numbness, and loss of strength [54]. In research in Saudi Arabia, neck and back pain were the main problems for dentists which could start to be corrected in the process of dental education. So, it is important for dental schools to improve dental ergonomics training for their students [55].

The main goal of dental ergonomics is to reduce the risk of MSDs and to minimize the amount of physical and mental stress so that the quality of dentists' work can be improved [56]. In addition, in the development of dental ergonomics research, the subject is not only dentists and dental students but should also extend to dental hygienists [57], dental assistants [58], and dental technicians [59]. The progress of dental ergonomics cannot be separated from its history, where initially dentists worked in a standing position; however, since the 1960s, the four-handed dentistry system has developed where dentists work in a sitting position [60]. Four-handed dentistry is a dental ergonomic effort to minimize unwanted movements and speed up dental treatment procedures [61].

Lietz et al., in their systematic review, reviewed various studies of ergonomic interventions to prevent MSDs in dental professionals. Of the 11 studies, 5 studies used ergonomic interventions in the form of using magnifying glasses or prismatic glasses; 2 studies used ergonomic dental chairs; 1 study used ergonomic dental instruments; and 3 studies provided ergonomic interventions in the form of ergonomic training. The results of all the included studies show the important role of ergonomic interventions that can provide improved work posture, increase work performance, and reduce the severity of MSDs in dental professionals [62]

  1. Reviewer comment:

In general, in this reviewer’s opinion, the strength of the manuscript is represented by the goal of the review, since the impact of human factors and ergonomics (HFE) in working activities is of paramount importance when dealing with occupational health and safety of medical professionals. Moreover, the bibliometric analysis could provide a knowledge mapping of the analyzed sector and   thus can represent a novelty.

Author's response and answer:

Page 2; Row: 52-68

 The most common MSDs in dentists are back pain, followed by neck pain, shoulder pain, high tension of the trapezius muscle, tendinitis, carpal tunnel syndrome, pinched nerves, early arthrosis, myopia, and auditive changes [8,9]. Pain in the muscles is an alarm in the body before the risk of paralysis and injury occurs, which has the potential to end a career early [10]. In their systematic review, Bret and Gorce reported that the highest prevalence of MSDs in dentists was in the lower back (>60%), shoulders, and upper extremities (35-55%). The main cause was an awkward posture repeated over a long time [11].  Soo et al. reported that dentists' susceptibility to MSDs reached 68% to 100% in various parts of the body, especially in the neck (26%-92%), shoulders (25%-92, 7%), and lower back (29% to 94.6%). The causes and problems of MSDs are multifactorial; several risk factors occur for female dentists (57.1%), with awkward posture (50%), with long periods of work (30%), and for specialist dentists (42.9%) [12]. To maximize the performance of dentists, human factors and ergonomics (HFE) is a special study area aimed at improving the health, safety, efficiency, and quality of dentists' work while also having a positive impact on patient safety. HFE interventions in health services are categorized into the following: (1) physical ergonomic interventions, (2) cognitive ergonomic interventions, and (3) organizational ergonomic interventions [13].

Page 17-18; Row: 424-428

In the author's visualization of keywords up to 2023, the keyword "Human Factors and Ergonomics (HFE)" was not found, even though HFE has progressed very rapidly recently. HFE is a science that studies interactions among humans, tasks, and elements of work systems, with the aim of making humans more integrated in a system via adaptations to the environment for the individual. Dentists and other dental health professionals will function better in a more conducive environment. HFE interventions have the potential to improve the performance, health, and welfare of workers, including (1) physical ergonomics interventions (anthropometrics, anatomy, physiology, biomechanics); (2) organizational ergonomics intervention (organizational structure, policies, procedures); and (3) cognitive ergonomic interventions relating to mental processes (memory, reasoning, perception, motor reactions) [68,69]. The research into the application of HFE in the dental education system and dental practice is a future challenge for the world.

  1. Reviewer comment:

Besides these positive aspects, several weaknesses must be outlined:

  • Research motivations are weak and the Authors need to elaborate more on the focus of the study and how it can contribute to augment knowledge in the field of

Author's response and answer:

Page  2; Row: 52-75

  • To maximize the performance of dentists, human factors and ergonomics (HFE) is a special study area aimed at improving the health, safety, efficiency, and quality of dentists' work while also having a positive impact on patient safety. HFE interventions in health services are categorized into the following: (1) physical ergonomic interventions, (2) cognitive ergonomic interventions, and (3) organizational ergonomic interventions [13].
  • Scientific publications on ergonomics related to dentists have been around since the 1960s and have consistently increased from year to year. The research subjects are not only dentists but also dental assistants [14], dental hygienists [15], and dental students [16]. The scope of the research is very broad, including the work environment [17,18] and ergonomic interventions [16,19–22]. These research articles have contributed to producing dental ergonomic principles so that dentists always work with an “ergonomic culture” [23] and provide strategies for preventing MSDs [24,25].
  • Based on this background, a bibliometric analysis was carried out on the topic of ergonomics and dentists. Bibliometric analysis is a statistical tool for mapping the highest and current levels of scientific development and identifying research gaps and trends for various purposes, such as searching for research opportunities and supporting scientific research [26].
  • The study design, methodology, and data analysis presented in the manuscript must be clarified and justified in a scientifically sound manner.

 Author's response and answer:

This has been stated in comment point 8.

Revision on Page 16; Row: 322-374

  • Research findings and practical implications of the study must be elaborated

Author's response and answer:

This has been stated in comment point 8.

Revision on Page 16; Row: 322-374

Revision on Page 17-18; Row: 424-435

In the author's visualization of keywords up to 2023, the keyword "Human Factors and Ergonomics (HFE)" was not found, even though HFE has progressed very rapidly recently. HFE is a science that studies interactions among humans, tasks, and elements of work systems, with the aim of making humans more integrated in a system via adaptations to the environment for the individual. Dentists and other dental health professionals will function better in a more conducive environment. HFE interventions have the potential to improve the performance, health, and welfare of workers, including (1) physical ergonomics interventions (anthropometrics, anatomy, physiology, biomechanics); (2) organizational ergonomics intervention (organizational structure, policies, procedures); and (3) cognitive ergonomic interventions relating to mental processes (memory, reasoning, perception, motor reactions) [68,69]. The research into the application of HFE in the dental education system and dental practice is a future challenge for the world.

  • Text proofreading is needed to improve language and eliminate some

Author's response and answer:

  • This manuscript has undergone English language editing by MDPI. The text has been checked for correct use of grammar and common technical terms, and edited to a level suitable for reporting research in a scholarly journal. MDPI uses experienced, native English speaking editors. Full details of the editing service can be found at

â–º https://www.mdpi.com/authors/english.

  1. Reviewer comment:

Comments on the Quality of English Language:

In the text some typos can be found. Hence, language proofreading is needed.

Author's response and answer:

  • This manuscript has undergone English language editing by MDPI. The text has been checked for correct use of grammar and common technical terms, and edited to a level suitable for reporting research in a scholarly journal. MDPI uses experienced, native English speaking editors. Full details of the editing service can be found at

â–º https://www.mdpi.com/authors/english.

Reviewer 3 Report

Comments and Suggestions for Authors

This research presents global trends in dental ergonomics publications using bibliometric analysis with VOSviewer software. The aim of this bibliometric analysis is to obtain quantitative and visual information on articles on dental ergonomics in Scopus-indexed publications from 1990 to 2023. The analysis of this research was performed on the basis of a performance analysis, an analysis of journals and articles, an analysis of collaboration between authors and between countries, and an analysis of the intellectual structure of authorship, which maps countries of publication, sources, authors, citation networks and co-citation networks between authors.

The bibliometric analysis presented is of interest to the community. The article follows a classic procedure for this type of study. The article is well structured, the methodological approach clear and the discussion pertinent. The objective is well defined and the hypotheses well formulated. However, the following minor comments and recommendations could further improve the article:

-          In the "introduction" section: Some references on the risks and prevalence of WMSD could be added. Some studies have proposed a meta-analysis of total prevalence and prevalence by body area, for health professionals or dentists worldwide, for example:

-          J Jacquier-Bret, P Gorce, Prevalence of body area work-related musculoskeletal disorders among healthcare professionals: a systematic review, International Journal of Environmental Research and Public Health 20 (1), 841

-          ZakerJafari HR, YektaKooshali MH.Work-Related Musculoskeletal Disorders in Iranian Dentists: A Systematic Review and Meta-analysis., Saf Health Work. 2018 Mar;9(1):1-9.

-          Soo SY, Ang WS, Chong CH, Tew IM, Yahya NA. Occupational ergonomics and related musculoskeletal disorders among dentists: A systematic review. Work. 2023;74(2):469-476.

These studies could be mentioned at the beginning of the introduction.

-          Why didn't the authors use the keyword "MSD" to search for articles in the scopus database, rather than “ergonom*”? Indeed, in the introduction, the authors present the study framework of the bibliometric analysis mainly around MSDs. Is there a reason for this?

-          Why did the authors study only the Scopus database? Why aren't the Science direct, Pubmed and Google Scholar databases used? Numerous journals addressing MSD issues can be found in these databasec. Can you explain and justify?

-          It may be interesting to add to the discussion the limits and constraints of bibliometric analysis.

Author Response

Dear Reviewer 3

This is my revision based on your review. I marked this revision with a gray highlighter.

Thank you for your attention.

  1. Reviewer comment:

This research presents global trends in dental ergonomics publications using bibliometric analysis with VOSviewer software. The aim of this bibliometric analysis is to obtain quantitative and visual information on articles on dental ergonomics in Scopus-indexed publications from 1990 to 2023. The analysis of this research was performed on the basis of a performance analysis, an analysis of journals and articles, an analysis of collaboration between authors and between countries, and an analysis of the intellectual structure of authorship, which maps countries of publication, sources, authors, citation networks and co-citation networks between authors.

Author's response and answer:

Thank you for the review

  1. Reviewer comment:

The bibliometric analysis presented is of interest to the community. The article follows a classic procedure for this type of study. The article is well structured, the methodological approach clear and the discussion pertinent. The objective is well defined and the hypotheses well formulated. However, the following minor comments and recommendations could further improve the article:

-     In the "introduction" section: Some references on the risks and prevalence of WMSD could be added. Some studies have proposed a meta-analysis of total prevalence and prevalence by body area, for health professionals or dentists worldwide, for example:

o    J Jacquier-Bret, P Gorce, Prevalence of body area work-related musculoskeletal disorders among healthcare professionals: a systematic review, International Journal of Environmental Research and Public Health 20 (1), 841

o    ZakerJafari HR, YektaKooshali MH.Work-Related Musculoskeletal Disorders in Iranian Dentists: A Systematic Review and Meta-analysis., Saf Health Work. 2018 Mar;9(1):1-9.

o    Soo SY, Ang WS, Chong CH, Tew IM, Yahya NA. Occupational ergonomics and related musculoskeletal disorders among dentists: A systematic review. Work. 2023;74(2):469-476.

 Author's response and answer:

Page 2; Rows: 56-63

In their systematic review, Bret and Gorce reported that the highest prevalence of MSDs in dentists was in the lower back (>60%), shoulders, and upper extremities (35-55%). The main cause was an awkward posture repeated over a long time [11].  Soo et al. reported that dentists' susceptibility to MSDs reached 68% to 100% in various parts of the body, especially in the neck (26%-92%), shoulders (25%-92, 7%), and lower back (29% to 94.6%). The causes and problems of MSDs are multifactorial; several risk factors occur for female dentists (57.1%), with awkward posture (50%), with long periods of work (30%), and for specialist dentists (42.9%) [12].

  1. Reviewer comment:

These studies could be mentioned at the beginning of the introduction.

-     Why didn't the authors use the keyword "MSD" to search for articles in the scopus database, rather than “ergonom*”? Indeed, in the introduction, the authors present the study framework of the bibliometric analysis mainly around MSDs. Is there a reason for this?

Author's response and answer:

Page: 3; Row: 98-111

This research was carried out in two stages. The first stage was an exploratory stage of searching for research topics using several keywords with Boolean expressions in the Scopus database. The purpose of this preliminary research was to find research topics with keywords that can provide the most data information. A preliminary research topic search was carried out on 5 April 2022 using Boolean sentences in the Scopus electronic database. To search for phrases in Scopus double quotes are used ("), wildcards (*), and Boolean operators (OR, AND, NOT). The purpose of double quotes is to tell Scopus that these are “loose phrases" meaning that the words must be together. The use of wildcards (*) to represent a number of characters and Boolean operators are used to expand or narrow search parameters when using databases or search engine. The default search field in Scopus uses TITLE-ABS-KEY because the Scopus database is an abstract indexer only [32,33].

 Table 1. Data mining with several topics in preliminary research

Boolean Search Sentences

Number of documents

(TITLE-ABS-KEY (ergonom*) AND TITLE-ABS-KEY (dentist*))

634

(TITLE-ABS-KEY (“musculoskeletal disorder*” OR ”MSDs*”) AND TITLE-ABS-KEY (dentist*))

372

(TITLE-ABS-KEY (“work fatigue*” OR ”burnout*”) AND TITLE-ABS-KEY (dentist*))

317

(TITLE-ABS-KEY (“work pressure*” OR ”work stress*” OR “work anxiety*”) AND TITLE-ABS-KEY (dentist*))

43

  1. Reviewer comment:

Why did the authors study only the Scopus database? Why aren't the Science direct, Pubmed and Google Scholar databases used? Numerous journals addressing MSD issues can be found in these databasec. Can you explain and justify?

  • Can be shifted at the beginning of the discussion

 Author's response and answer:

Page: 15; Row: 310-321

 The Scopus database was used as the source of bibliographic data in this research because it has wide coverage, has good data quality and accuracy, provides various bibliometric analysis features, and is a data source that is verified and academically recognized [42]. Falagas et al. compared the strengths and weaknesses of PubMed, Scopus, Web of Science, and Google Scholar. PubMed and Google Scholar are free, while PubMed is optimal for biomedical research, but Google Scholar’s accuracy is inconsistent. Scopus has 20% greater citation analysis coverage compared to Web of Science [43]. Sing et al. compared three Web of Science databases, Scopus and Dimensions. It was reported that almost all journals on the Web of Science are to be found in Scopus and Dimensions. Scopus indexes 66.07% more unique journals compared to Web of Science. Web of Science and Scopus coverage tends to be in the areas of life sciences, physical sciences, and technology, but Dimensions covers more social sciences and arts and humanities [44].

  1. Reviewer comment:

It may be interesting to add to the discussion the limits and constraints of bibliometric analysis.

 Author's response and answer:

Page: 18; Row: 438-446

 There are several limitations in bibliometric analysis, namely that open access to scientometric data is required. Access to data with sufficient accuracy is fundamental in bibliometric analysis. The important information required includes metadata, author data, affiliations, and citations. Another limitation is the possibility that the downloaded data are incomplete or duplicate data. The main obstacle in bibliometric analysis is the complexity and diversity of bibliographic data, so researchers need to be careful in understanding the various dimensions of the data. The number of citations is directly proportional to time, meaning that older papers tend to receive more citations than new papers.

Round 2

Reviewer 2 Report

Comments and Suggestions for Authors

The Authors have satisfactorily improved the quality of the manuscript. Hence, it can be considered for publication.